

# Wood identification based on macroscopic images using deep and transfer learning approaches

Halime Ergun

Department of Computer Engineering, Seydişehir Ahmet Cengiz Faculty of Engineering, Necmettin Erbakan University, Konya, Turkey

## ABSTRACT

Identifying forest types is vital for evaluating the ecological, economic, and social benefits provided by forests, and for protecting, managing, and sustaining them. Although traditionally based on expert observation, recent developments have increased the use of technologies such as artificial intelligence (AI). The use of advanced methods such as deep learning will make forest species recognition faster and easier. In this study, the deep network models RestNet18, GoogLeNet, VGG19, Inceptionv3, MobileNetv2, DenseNet201, InceptionResNetv2, EfficientNet and ShuffleNet, which were pre-trained with ImageNet dataset, were adapted to a new dataset. In this adaptation, transfer learning method is used. These models have different architectures that allow a wide range of performance evaluation. The performance of the model was evaluated by accuracy, recall, precision, F1-score, specificity and Matthews correlation coefficient. ShuffleNet was proposed as a lightweight network model that achieves high performance with low computational power and resource requirements. This model was an efficient model with an accuracy close to other models with customisation. This study reveals that deep network models are an effective tool in the field of forest species recognition. This study makes an important contribution to the conservation and management of forests.

## INTRODUCTION

The identification and classification of wood species serve as crucial steps towards understanding their biological diversity, ecological role, economic value, and cultural significance. Accurate identification is also necessary for the appropriate use of wood (*Wheeler & Baas, 1998*). While identifying the species is relatively straightforward when organs such as flowers, leaves, or seeds are present, it becomes a challenging task once the wood is processed. The identification relies solely on the macroscopic or microscopic characteristics of the wood (*Khalid et al., 2008*). Macroscopic structure refers to the features observable to the naked eye or under low magnification, while microscopic structure denotes the cell-level characteristics observable under high magnification.

Corresponding author
Halime Ergun, hboztoprak@erbakan.edu.tr

Sometimes, identification at the genus level is sufficient for diagnosing the type of wood. However, there are instances where species-level identification is necessary. For instance, while trade for certain wood species may be banned or restricted, other species within the same genus may be freely traded (such as *Dalbergia* spp., *Pterocarpus* spp.) (*Gasson, 2011*; *Shou et al., 2014*; *Snel et al., 2018*). Furthermore, species-level identification is needed when gathering information about structures of cultural and historical value (*Hwang et al., 2016*). Microscopic methods are preferred in such cases; however, some wood species cannot be distinguished from each other microscopically (*Tuncer, 2020*). Therefore, the work on improving methods for diagnosing wood species continues, with image processing techniques showing significant advancements due to emerging technologies.

Traditional methods for diagnosing the type of wood involve examining its macroscopic or microscopic structure. The microscopic technique examines the wood's anatomy using light or electron microscopy. It identifies cell types, measures cell wall thickness, determines cell sizes and shapes, analyzes cell contents, and examines the wood's cell wall chemistry. The macroscopic method observes the wood's physical and anatomical features with the naked eye or a hand magnifier. It identifies properties such as color, density, hardness, odor, grain, fibers, pores, luster, texture, resin, grain structure, cracks, knots, and more. This reveals the wood's natural and unique qualities, which can vary depending on factors like age, growth conditions, cutting time, drying method, processing, decay level, and staining or varnishing. Although these methods are considered the most reliable, they are time-consuming, labor-intensive, and require expertise in wood anatomy. As a result, there is a growing demand for new and different diagnostic methods (*Dormontt et al., 2015*). In an era where quick and easy access to information is expected, new technologies need to be developed for industries working with wood materials and for maintaining environmental balance.

The limitations of these traditional methods underscore the need for more practical, faster, and cheaper alternatives in diagnosing wood types. One such alternative involves computer vision techniques compatible with widely used devices like smartphones. Consequently, various studies have been conducted in the area of wood type recognition and classification using computer vision techniques (*Martins et al., 2013*; *Yadav et al., 2013*, *2015*; *Rosa Da Silva et al., 2017*; *Fabijańska, Danek & Barniak, 2021*). These studies have used macroscopic or microscopic images of wood samples. Macroscopic images are 10–15× magnified photographs of wood surfaces, while microscopic images are photographs of wood samples magnified between 25–100× under a microscope. The species of wood are identified by applying feature extraction and classification processes on these images. The drawback of these methods, however, is the need for specialized equipment to capture the images (*Filho et al., 2014*).

In the process of wood species recognition through image processing, a two-stage model is typically employed: feature extraction and classification. During the feature extraction phase, a variety of methods can be utilized to convert images into numerical data. One such method is the Gray Level Co-occurrence Matrices (GLCM). This technique calculates pixel gray level differences that reflect the texture properties of the image. The features

derived from the image are then used as input in the classification stage (*Khalid et al., 2008*; *Piuri & Scotti, 2010*; *Mallik et al., 2011*; *Martins et al., 2013*; *Yadav et al., 2013*).

In the classification phase, wood samples are trained using different algorithms, and a statistical model is built. This model classifies new examples according to wood species. Among these algorithms are multilayer artificial neural networks (*Yusof & Rosli, 2013*; *Zhao, Dou & Chen, 2014*; *Ibrahim et al., 2018*) and support vector machines (*Turhan & Serdar, 2013*; *Martins et al., 2013*; *Filho et al., 2014*; *Barmpoutis et al., 2018*), and methods like KNN. These methods have their pros and cons, and their performance depends on the dataset and feature extraction method (*Fabijańska, Danek & Barniak, 2021*).

However, traditional methods for identifying and classifying wood species often require examination of the wood's anatomical structure, which can be time-consuming and require expertise. Deep learning, on the other hand, employs artificial neural networks that learn from and make predictions based on data. Thus, they can analyze data, learn from it, and make predictions. As a result, deep learning techniques are yielding faster and more effective results in this field. These techniques can also learn to identify new types of wood. The model can be updated with new data and cover a broader range of wood species. However, deep learning techniques require a substantial amount of data for accurate wood type diagnosis. To make accurate predictions, images of a wide variety of wood types are needed. Gathering and labeling this data can be challenging. Deep learning techniques can be affected by factors like image quality, lighting, angle, background, *etc*. These factors can lead to image degradation or misinterpretation.

*Sun et al. (2021)* developed a transfer learning approach to wood species identification using a limited data set. Linear discriminant analysis (LDA) and k-nearest neighbours (KNN) were used as classification methods to enhance the features obtained from the ResNet50 model (*Sun et al., 2021*).

However, *Fabijańska, Danek & Barniak (2021)*, using a macroscopic data set of 14 species of corals and angiosperms common in Europe, developed a method that now uses an evolutionary encoder network and a sliding window layout. This study shows that a method that now uses a sliding window layout with an evolutionary encoder network can diagnose wood species with high accuracy, regardless of the spatial resolution of the images.

A study by *Kırbaş & Çifci (2022)* compared the transfer learning performance of various deep learning architectures on a WOOD-AUTH dataset comprising 12 wood species. The study evaluated the transfer learning performance of ResNet50, Inceptionv3, Xception, and VGG19. The results show that the Xception model achieves the highest accuracy (*Kırbaş & Çifci, 2022*). In a related study, researchers proposed a wood recognition approach utilizing ResNet50 to extract the textile properties of wood, applying global average pooling (GAP) to decrease attribute numbers and boost model generalizability. Extreme learning machine (ELM) algorithm was then implemented for classification. ELM is a one-layer advanced neural network capable of learning properties from images without adjusting the parameters of hidden layer or backward spread.

*Elmas (2021)* applied pre-trained convolutional neural networks such as AlexNet, DenseNet201, ResNet18, ResNet50, ResNet101, VGG16, and VGG19 on a data set
consisting of 24,686 wood bark images belonging to 59 wood species collected from various regions of Turkey using the transfer learning method, and stated that the highest accuracy value was obtained with DenseNet121 (*Elmas, 2021*). *Miao et al. (2022)* proposed a convolutional neural network developed for wood type recognition from wood images in their study. The model combines Inception and mobilenetV3 networks. The model was tested with a large-scale data set containing 16 different wood types. The model achieved 96.4% accuracy (*Miao et al., 2022*). *Wu et al. (2021)* used deep learning methods for wood identification based on longitudinal cross-section images in their study. The model benefited from effective convolutional neural networks such as ResNet50, DenseNet121 and MobileNetv2. The model was evaluated with a data set containing 11 common wood types. The model achieved 98.2% accuracy (*Wu et al., 2021*). These studies reveal that deep learning techniques are more successful than traditional methods in the field of wood type identification and classification.

There are some specific challenges in identifying wood species. The anatomical structure of the species, the prevailing climate and soil conditions, forest density in the region, the amount and direction of sunlight received, among other factors, can affect the characteristics and classification of wood species. Another challenge is that large, public data sets related to wood images are only available for a specific region. The transfer learning method allows pre-trained convolutional neural networks to be used to extract features from wood images. This reduces the need for more data to train the model and increases the model's generalization ability.

In this study, a transfer learning method was used that fine-tuned pre-trained convolutional neural networks (CNN) models such as ResNet50, DenseNet121, MobileNetv2 and ShuffleNetv2. This method transfers the information learned from ImageNet, a comprehensive data set consisting of natural images, to our task of determining wood types using macroscopic images. Most of the state-of-the-art models examined did not report on different performance metrics that help gain more insight into the proposed model and solution approach. The structure of CNN models is becoming increasingly complex, hardware costs are rising and application efficiency is decreasing, therefore lightweight deep learning model research is an inevitable trend. An attempt was made to achieve a success close to more complex models by using a customized ShuffleNet architecture with different parameters in the recognition of wood species.

## MATERIALS AND METHODS

Recognition of wood species typically demands a cross-sectional review of wood features by experts. These characteristics include vessel arrangement, pore direction, ray parenchyma, fiber, and other surface information. The identification standards from the IAWA list (*Wheeler & Baas, 1998*) are used to compare these features between wood species (*Sun et al., 2021*).

### Dataset

A dataset of macroscopic images was used for forest species detection and classification (*Cano Saenz et al., 2022*). Ten species were chosen for analysis, ensuring a minimum of 800

**Table 1 The species and image numbers used in this study.**

|  | Tür | Train | Test | Image number |
|---|---|---|---|---|
| 1 | *Achapo* | 800 | 389 | 1,189 |
| 2 | *Cedro costeno* | 800 | 328 | 1,128 |
| 3 | *Chanul* | 800 | 201 | 1,001 |
| 4 | *Cipres* | 800 | 15 | 815 |
| 5 | *Cuangare* | 800 | 300 | 1,100 |
| 6 | *Eucalipto blanco* | 800 | 305 | 1,105 |
| 7 | *Guayacan amarillo* | 800 | 306 | 1,106 |
| 8 | *Nogal cafetero* | 800 | 129 | 929 |
| 9 | *Sajo* | 800 | 23 | 823 |
| 10 | *Urapan* | 800 | 225 | 1,025 |
|  | Total | 8,000 | 2,221 | 10,792 |

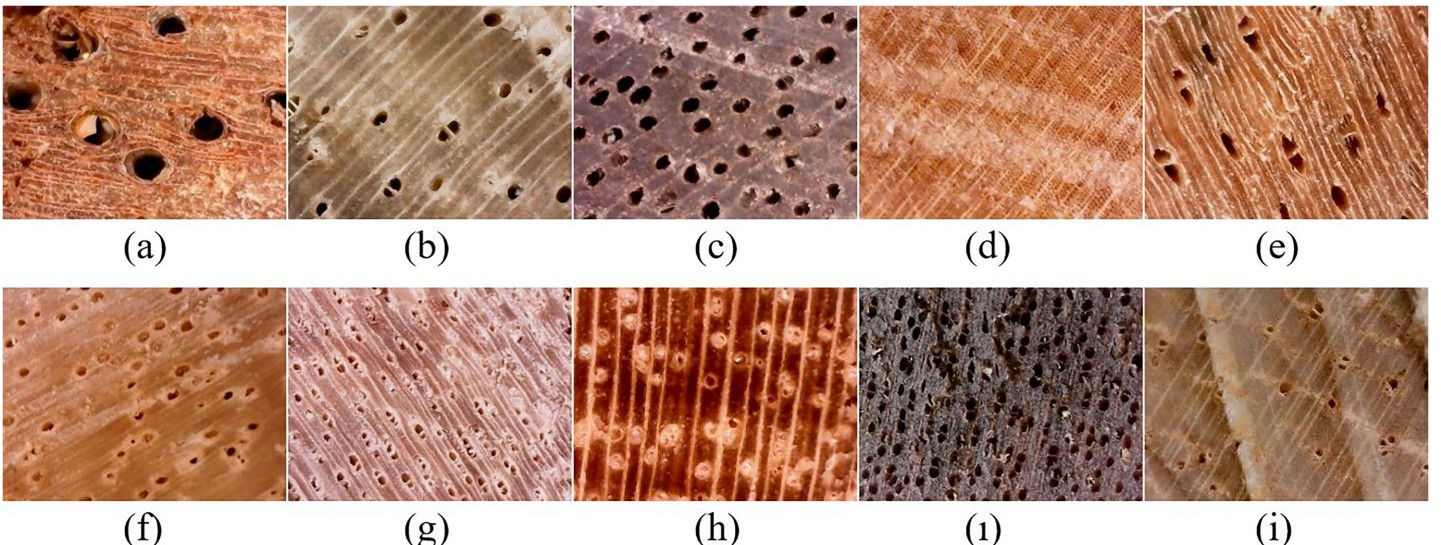

**Figure 1 Macroscopic images of sample species.** (A) *Achapo*, (B) *Cedro Costeno*, (C) *Chanul*, (D) *Cipres*, (E) *Cuangare*, (F) *Eucalipto Blanco*, (G) *Guayacan Amarillo*, (H) *Nogal Cafetero*, (ı) *Sajo*, (I) *Urapan* (*Cano Saenz et al., 2022*).

images for each category. In total, the dataset comprised 10,792 images of tropical forest species. The training module employed a subset of 8,000 images chosen at random, while the test group incorporated a maximum of 2,221 images. This equal allocation of the training data will facilitate a more precise assessment of the model's overall performance. Detailed information regarding the types employed and the total number of images can be found in Table 1.

The data set contains macroscopic images obtained using a scale that allows the observation of pores, fibers and parenchyma. The data set includes more than 8,000 images with a resolution of 640 × 480 pixels, containing 3.9 microns per pixel, and an area of (2.5 × 1.9) square millimeters where anatomical features are displayed. Figure 1 shows macroscopic images of species.

## Transfer learning

Transfer learning is a technique that allows a machine learning model to adapt and use the information it learned for one task for another task. CNN models are layered artificial neural networks that extract complex features from images. Training CNN models from scratch can be difficult and time consuming. Therefore, using pre-trained networks, that is, networks with previously calculated weights on a large data set (such as ImageNet), is more common (*Abu Al-Haija, 2022*; *Elmas, 2021*; *Katsigiannis et al., 2023*).

Fine-tuning CNN is a transfer learning method that allows a machine learning model to adapt and use the information it learned for one task for another task. Instead of starting the CNN's weights with random values, it makes more sense to use a pre-trained weight set. This weight set provides a better starting point than random values, even if it comes from a different problem. Changing the last layer (classifier) and retraining enabled the fine-tuning of the weights of the CNN with backpropagation. In addition, this approach allows us to choose how many layers will be fine-tuned. Usually, the first layers are basic filters that detect low-level general features such as edges and colours. These filters are useful for most image classification tasks (*De Geus et al., 2021*; *Sanida et al., 2023*). The most common type of transfer learning is to replace the fully connected layer and transfer all layers.

Making wood categorization from macroscopic images is distinct from the problem in the ImageNet data set. In this study, networks trained for 1,000 classes were fine-tuned for 10 classes. The fine-tuning process is carried out separately for each model. Each model was launched using its weights trained with the ImageNet data set. The last layer of the model has been replaced with 10 classes output layer. The weights of this layer are initiated randomly. During the fine adjustment procedure, all layers of the model have continued to be trained. Thus, the classification layers of the networks have been re-trained for 10 classes.

The convolutional layers of the network extract various features from the input image. The last learnable layer and the last classification layer use these features to classify the image. To retrain a pre-trained network to classify new images, these two layers are redesigned to be suitable for the new data set. Replacing these two layers is an important step to adapt a pre-trained network to a new task. However, this is not enough. In addition, the weights of the other layers of the network need to be fine-tuned. This allows the network to better fit the new data set and achieve higher accuracy. The network is fine-tuned by training the last layers with or without freezing them. This adapts the network's higher-level features to the new task. Alternatively, all layers of the network can be trained, thus updating both low- and high-level features of the network. However, this approach requires more computational power and data (*Katsigiannis et al., 2023*).

The last layer with learnable weights is usually a fully connected layer. This layer is replaced with a new layer that has the same number of outputs as the number of classes in the new data set. For some networks, such as SqueezeNet, the last layer with learnable weights is a 1-by-1 convolutional layer. In this case, it is replaced with a new convolutional layer that has the same number of filters as the number of classes. In this study, different

deep learning architectures that achieve successful results for various image classification tasks are used. These include: layered residual networks (ResNet), a complex image recognition system based on convolutional neural networks (Inceptionv3), a crossbreed architecture for deep convolutional neural networks (Xception), a multi-layer visual perception network (VGGNet), image recognition and classification with deep convolutional neural networks (GoogLeNet), a lightweight and efficient convolutional neural network for mobile devices (MobileNetv2), densely connected convolutional networks (DenseNet201), and a convolutional neural network that combines Inception modules with residual connections (InceptionResNetv2). The models are:

- **VGGNet:** A deep learning model with 41 or 47 layers that uses very small (3 × 3) convolution filters. It consists of 13 or 16 convolution layers, three fully connected layers, pooling, activation function, dropout and classification layer.

- **ResNet:** A deep learning model with 18, 50 or more layers that uses a technique called residual learning. This technique allows the network to be deeper and prevents gradient vanishing. The ResNet model consists of many residual blocks that are connected to each other. A residual block adds its own output to the input data and passes it to the next layer (*He et al., 2016*).

- **GoogLeNet:** A CNN model with 22 layers that uses inception modules to increase the network depth and width while reducing the parameters (*Szegedy et al., 2015*).

- **MobileNetv2:** A CNN model with 88 layers that uses inverted residual blocks with linear bottlenecks to reduce the computational cost and increase the accuracy (*Sandler et al., 2018*).

- **DenseNet201:** A CNN model that uses dense connections to increase the information flow and parameter efficiency (*Huang et al., 2017*). This structure ensures that each layer is connected to all previous layers.

- **InceptionResNetv2:** A convolutional neural network that combines inception modules with residual connections. It is a deep neural network with 164 layers. It offers the advantages of both inception and ResNet techniques (*Szegedy et al., 2017*).

- **Inception:** Inception is based on applying filtering and pooling operations simultaneously in convolutional layers. It works modularly. InceptionV3 is an improved version of GoogLeNet with 48 layers that uses factorized convolutions and label smoothing to increase performance (*Szegedy et al., 2016*).

- **Xception:** Xception network offers two different approaches, depthwise convolution and pointwise convolution, in addition to the improvements in Inception V3 (*Chollet, 2017*).

- **ShuffleNet:** ShuffleNet is a type of convolutional neural network that is very efficient and fast for mobile devices. It uses two new techniques, pointwise group convolution and channel shuffle, to reduce the computation and memory requirements while maintaining high accuracy (*Zhang et al., 2018*). ShuffleNetv2 is a CNN model with 50 layers that uses channel shuffle and split operations to increase the speed and performance (*Ma et al., 2018*).

**EfficientNet:** EfficientNet is a CNN model that uses compound scaling to balance the network depth, width and resolution, which provides better efficiency and accuracy (*Tan & Le, 2019*). B0 to B7 are different variants of EfficientNet with different scales and parameters; where B0 is the base model and B7 is the largest and most complex model.

## Proposed method

ShuffleNet is a convolutional neural network optimized for speed and memory. ShuffleNet, a lightweight CNN architecture, has convolutional layers, group convolutional layers, depthwise convolutional layers, channel shuffle layers, pooling layers and fully connected (FC) layers in its network structure (*Zhang et al., 2018*). Group convolution layer performs the operation of using different convolution filter groups on the same input data. Depthwise convolutional layers have less parameters than convolutional layers. This reduces overfitting in the model and makes it a useful model for mobile applications. Channel shuffle layer reshapes the output channel size as (g,c) with g group convolution layer and gxc channels inside, transposes it and flattens it as the input of the next layer. Pooling layer reduces the input size while preserving the perceived features (*Zhang et al., 2018*; *Toğaçar & Ergen, 2022*). Channel shuffle increases the information flow by shuffling the channels in different groups. Thanks to these techniques, ShuffleNet achieves similar performance with more complex models while requiring less parameters and computation power.

The choice of an architecture for transfer learning depends on various factors such as the size and similarity of the source and target data sets, the complexity and accuracy of the pre-trained model, and the available computation resources. For example, a complex and accurate model such as ResNet50 or DenseNet121 can be chosen if the target data set is small and similar to the source data set, and most of the layers can be frozen except the last ones. A simple and efficient model such as MobileNetv2 or ShuffleNetv2 can be chosen if the target data set is large and different from the source data set, and most of the layers can be **fine**-tuned or even trained from scratch. A light and fast model such as ShuffleNetv2 or modified ShuffleNet that can achieve similar performance to more complex models while requiring less memory and power can be chosen if the computation resources are limited.

Some modifications were made to the ShuffleNetv2 architecture and it was retrained. The last layer of ShuffleNetv2 was changed. Instead of $1 \times 1$ convolution, global average pooling and fully connected layer, the last layer used $1 \times 1$ convolution, global max pooling and fully connected layer. The reason for this was that max pooling had more discriminative power than average pooling. Max pooling selects the highest value from each feature map, while average pooling takes the average of each feature map (*Ma et al., 2018*). In this way, max pooling preserves important features while discarding noise and unnecessary information. This means that the maximum pooling emphasizes certain features, regardless of the location of the properties. The average pooling, however, preserves the essence of the features and gives a smoother appearance. This suggests that the difference between average pooling and maximum pooling may vary depending on the data set, network architecture, and training parameters.

Leaky ReLU was used as the activation function. Such a modification was stated to help the model prevent dead neurons and learn better (*Ma et al., 2018*). A dead neuron is a neuron that has zero output for any input and therefore cannot learn anything. This happens when the neuron's weights are negative and the gradient of ReLU is zero. As a result, the neuron cannot be updated by backpropagation and remains dead.

To improve the learning ability of the model for partial feature values (<0), it is suggested to apply LeakyReLU activation function to the construction process of Shuffle Net lightweight convolutional neural network (*Song, Zhang & Long, 2023*). The expression of LeakyReLU activation function is given in Eq. (1).

$$LeakyReLU = \begin{cases} a_i x, & x < 0 \\ x, x \geq 0 \end{cases}. \tag{1}$$

Here, x is the input eigenvalue, $a \in (0, 1)$ and when $x < 0$, the output value of the LeakyReLU activation function is a x. When $x \geq 0$, the output value of the LeakyReLU activation function is x.

## Evaluation metrics

In this section, various metrics are used to measure the performance of our models on our data set. The metrics show how accurate, sensitive, specific, precise and balanced the model is. They also measure how well the model predicts better than random guessing. These metrics are based on the concepts of true positives (TP), false positives (FP), false negatives (FN) and true negatives (TN). TP are the samples that are correctly classified as positive by the model; FP are the samples that are incorrectly classified as positive by the model; FN are the samples that are incorrectly classified as negative by the model; and TN are the samples that are correctly classified as negative by the model (*Toğaçar & Ergen, 2022*; *Tharwat, 2020*).

**Accuracy:** The ratio of the images that the model predicted correctly to the total images.

**Error:** The ratio of the images that the model predicted incorrectly to the total images. Accuracy and Error show the overall performance of the model, but they do not reflect the imbalance between the classes. For example, if one class has much more samples than the other, the model can achieve high Accuracy by predicting that class more often, but it can ignore the other classes.

$$Global\ Accuracy = \frac{TP + TN}{TP + TN + FP + FN}. \tag{2}$$

**Recall:** The ratio of the images that the model predicted correctly to the images that belong to the actual class. Precision: The ratio of the images that the model predicted correctly to the images that the model assigned to that class.

$$precision = \frac{TP}{TP + FP}, \qquad recall = \frac{TP}{TP + FN}. \tag{3}$$

**F1-score:** Score indicates how well the projected boundary of each class is aligned with the actual boundary.

$$\text{F1-score} = \frac{2 * precision * recall}{recall + precision}, \quad (4)$$

**Specificity:** The ratio of the images that the model predicted correctly to the images that do not belong to the actual class.

$$Specificity = \frac{TN}{TN + FP}. \quad (5)$$

**MCC:** Matthews correlation coefficient. It takes values between −1 and +1. +1 means perfect prediction, 0 means random prediction and −1 means inverse prediction.

$$MCC = \frac{TP * TN - FP * FN}{\sqrt{(TP + FP)(TP + FN)(TN + FP)(TN + FN)}}. \quad (6)$$

The popular metrics mentioned above (accuracy, recall, precision, *etc.*) do not fully provide a result for data sets that are unevenly distributed. Basically, it evaluates by looking at the correlation (phi-coefficient) relationship between real data and predicted data. MCC criterion, which uses all the parameters in the confusion matrix as seen in Eq. (6).

## EXPERIMENTAL RESULTS

In this study, RestNet18, GoogLeNet, VGG19, Inceptionv3, MobileNetv2, DenseNet201, InceptionResNetv2, EfficientNetb0, ShuffleNet deep learning models were used for transfer learning. These models were trained for a general-purpose visual recognition task and learned high-level features to recognize different objects. The last layer of these models was removed and a new layer with neurons equal to the number of forest species was added. This new layer was initialized with random weights and trained with the forest species data set. Thus, the model adapted the general features learned in the source domain to the target domain.

The models in Table 2 are artificial neural networks that are used for visual processing tasks such as image classification. The size and number of parameters of each model indicate the complexity and efficiency of the model. The number of parameters specifies the number of weights that the model can learn.

The classification accuracies of the fine-tuned models on the test images were calculated (Table 3). No layer was frozen during the model execution. Data augmentation was not applied. The mini batch size was set to 64. The learning rate was chosen low because transfer learning does not require training for many epochs. Retraining updates the network to learn and identify features associated with new images and categories. In most cases, retraining requires less data than training a model from scratch. After the model was retrained, a total of 2,221 test images were classified and the performance of the networks was evaluated.

Different metrics can be used to compare the performance of image classification models. To determine the model that shows the best performance according to any of these metrics, it is necessary to train and test different models on the same data set. In this way, it

**Table 2 The details of CNN models used in this study.**

| Model neural network | Depth | Size | Parameters (Millions) |
|---|---|---|---|
| GoogLeNet | 22 | 27 MB | 7.0 |
| Inceptionv3 | 48 | 89 MB | 23.9 |
| DenseNet201 | 201 | 77 MB | 20.0 |
| MobileNetv2 | 53 | 13 MB | 3.5 |
| RestNet18 | 18 | 44 MB | 11.7 |
| EfficientNetb0 | 82 | 20 MB | 5.3 |
| InceptionresNetv2 | 164 | 209 MB | 55.9 |
| ShuffleNet | 50 | 5.4 MB | 1.4 |
| VGG19 | 19 | 535 MB | 144.0 |

**Table 3 The classification performance of CNN models used in this study.**

| Models | Accuracy | Error | Recall | Specificity | Precision | F1-score | MCC |
|---|---|---|---|---|---|---|---|
| RestNet18 | 0.9568 | 0.0432 | 0.9675 | 0.9951 | 0.9382 | 0.9511 | 0.9471 |
| GoogLeNet | 0.9216 | 0.0784 | 0.9348 | 0.9912 | 0.8929 | 0.9061 | 0.9016 |
| VGG19 | 0.8676 | 0.1324 | 0.9014 | 0.9851 | 0.8160 | 0.8361 | 0.8331 |
| Inceptionv3 | 0.9572 | 0.0428 | 0.9672 | 0.9952 | 0.9234 | 0.9420 | 0.9389 |
| MobileNetv2 | 0.9514 | 0.0486 | 0.9630 | 0.9945 | 0.9160 | 0.9356 | 0.9321 |
| DenseNet201 | 0.9698 | 0.0302 | 0.9770 | 0.9966 | 0.9483 | 0.9607 | 0.9583 |
| InceptionResNetv2 | 0.9734 | 0.0266 | 0.9734 | 0.9970 | 0.9606 | 0.9667 | 0.9638 |
| EfficientNetb0 | 0.9644 | 0.0356 | 0.9720 | 0.9960 | 0.9261 | 0.9456 | 0.9433 |
| ShuffleNet | 0.9356 | 0.0644 | 0.9481 | 0.9927 | 0.8944 | 0.9161 | 0.9114 |
| Modified ShuffleNet | 0.9604 | 0.0308 | 0.9701 | 0.9906 | 0.9413 | 0.9527 | 0.9503 |

is possible to compare the results of the models objectively. All these parameter values are obtained for the same database and compared.

The modified ShuffleNet requires more training than other models because some weights and biases are not transferred. Table 1 shows the effect of the changes made on the model performance. Figure 2 presents the confusion matrix for the test images of the ShuffleNet model that was changed with the proposed method. The confusion matrix is a table that summarizes the prediction results for a classification problem. The confusion matrix allows us to get an idea about the types that are misclassified by the classifier. The total accuracy of the model is 92%, which means that it correctly predicts 92% of the test data.

The model has a high accuracy for classes 1, 2, 3, 6 and 7, which are *Achapo, Cedro costeno, Chanul, Eucalipto Blanco* and *Guayacan Amarillo*. The accuracy of these classes is above 95% and there are very few misclassifications. The model has a low accuracy for class 4, which is *Cipres*. This class has only 71% accuracy and many misclassifications. Species-level identification of wood types is more difficult than genus-level, because some

**Figure 2 Confusion matrix for ShuffleNet model.**

| | Achapo | Cedro Costeno | Chanul | Cipres | Cuangare | Eucalipto Blanco | Guayacan Amarillo | Nogal Cafetero | Sajo | Urapan |
|---|---|---|---|---|---|---|---|---|---|---|
| Achapo | 375 | 5 | | 2 | 2 | 1 | | 3 | | |
| Cedro Costeno | 2 | 309 | 2 | 1 | 5 | | | 3 | 1 | 5 |
| Chanul | | | 198 | | 1 | | | 2 | | |
| Cipres | | | | 15 | | | | | | |
| Cuangare | | 5 | | | 293 | | | | 1 | 1 |
| Eucalipto Blanco | | | | 1 | 2 | 288 | 7 | | | 7 |
| Guayacan Amarillo | | | | | 5 | | 297 | 3 | 1 | |
| Nogal Cafetero | | 1 | 1 | 2 | 1 | | | 124 | | |
| Sajo | | | | | | | | | 23 | |
| Urapan | | 5 | | | | 1 | | | | 219 |

wood types within the same genus can have very similar macroscopic features that are hard to distinguish by human eye or computer vision techniques (*Schweingruber & Baas, 2011*).

## DISCUSSION

Several pre-trained CNN models such as ResNet50, DenseNet121, MobileNetv2 and ShuffleNetv2 were fine-tuned using a transfer learning method in this study. This method enables the transfer of the knowledge learned from ImageNet, which is a large and diverse data set of natural images, to the task of identifying wood types using macroscopic images (*Abu Al-Haija, 2022*). Other transfer learning methods that can be applied for wood species identification are feature extraction, where pre-trained models are used as feature extractors and a new classifier is trained on top of them; or multi-task learning, where the pre-trained models are trained on multiple related tasks simultaneously (*Elmas, 2021*). The weights of the previous layers in the network can be optionally frozen by setting the learning rates in those layers to zero (*Katsigiannis et al., 2023*). The parameters of the frozen layers are not updated during training, which can significantly speed up the network training. If the new data set is small, freezing weights can also prevent the network from overfitting to the new data set. In general, if the data set is small or similar, fewer layers are frozen or fewer weights are changed. If the data set is large or different, more layers are frozen or more weights are changed. If the target data set is large, this means that there is enough data to learn and avoid overfitting. Therefore, freezing more layers can speed up

the training process and reduce the computation cost without sacrificing much accuracy. It can also prevent the model from forgetting the useful features learned from the source data set; changing more weights can help the model adapt to the new domain (*Sanida et al., 2023*).

There are many pre-trained models available for transfer learning, and each has its advantages and disadvantages that need to be considered: When choosing these models, factors such as size, accuracy and prediction speed should be taken into account. Size, the importance of your model size will vary depending on where and how you plan to use it. Accuracy, shows the generalization ability of the model. However, a low accuracy score on ImageNet does not mean that the model will perform poorly on all tasks. Prediction speed determines the performance of the model. These factors can vary depending on the architecture of the model and the data set it was trained on. The model structure of CNNs is becoming more and more complex, the hardware cost is increasing and the application efficiency is decreasing, so lightweight deep learning model research is an inevitable trend (*Song, Zhang & Long, 2023*).

Lightweight and computationally efficient models; MobileNetv2 and ShuffleNet are good options when there are limitations on the model size. MobileNetV2 is good for reducing network size and cost but contains subsampling bottlenecks (*Pradipkumar & Alagumalai, 2023*). ShuffleNetv2 is a convolutional neural network optimized for speed and memory. ShuffleNet has been reported to achieve ~13X real speed increase compared to AlexNet CNN while maintaining similar accuracy levels (*Abu Al-Haija, 2022*). There are studies that place the compression and excitation blocks after each stage of the Shuffle Block to reduce the computation cost. New architectures based on ShuffleNets are being designed (*Tang et al., 2020*; *Pradipkumar & Alagumalai, 2023*).

ShuffleNet is a good choice for wood species identification because it is designed to reduce the computational complexity and increase the efficiency of convolutional neural networks. It uses two techniques called group convolution and channel shuffle to achieve this goal. These techniques reduce the number of parameters and operations while increasing the feature diversity and information flow of the model. ShuffleNet has potential advantages over other models such as mobileNetv2 for wood species identification. ShuffleNet is faster and more accurate than MobileNetv2 in various tasks such as image classification, object detection and face recognition (*Zhang et al., 2018*). Moreover, as our experimental results also show, ShuffleNet can adapt better to different data sets and transfer learning methods.

The limitations and challenges of this study are that collecting, processing and labelling wood images is a laborious and time-consuming process. Also, situations such as the same wood species looking different in different regions or different wood species resembling each other can make identification difficult. There is no standard data set or evaluation metric for wood species identification with deep learning. Therefore, it is hard to compare or generalize the results of different studies. Most of the studies have used regional data bases and focused on a limited number of species. There is a need for a more general reference database (*Ergun & Uzun, 2022*).

## CONCLUSIONS

In this study, different deep learning architectures were used to identify wood species. The architectures used include ResNet, Inceptionv3, Xception, VGGNet, GoogLeNet, MobileNetv2, DenseNet201 and ShuffleNet. Metrics such as accuracy, sensitivity, precision, F1-score, specificity and MCC were used to evaluate the performance. Deeper and more complex models performed better. However, these models required more computation power and time. The modified ShuffleNet reduces complexity and computation costs while achieving the same level of success.

Machine and deep learning are powerful techniques that can be used to identify wood species. However, these techniques need lightweight networks to work on mobile devices. The results show that the modified ShuffleNet model achieves similar performance to more complex models while requiring less computation power, making it suitable for mobile applications. The study demonstrates the potential of deep learning techniques for identifying wood types using macroscopic images. It also provides a benchmark for future research in this field. In the next studies, feature extraction and transfer learning methods can be compared; different types of data sets can be used, the degrees of similarity or difference between different wood species can be examined; a more general reference data set can be created; more user-friendly and accessible applications and tools can be designed for wood species identification.

In future studies, it may be possible to improve the performance of wood species identification by using different feature extraction methods and different transfer learning methods (multi-task learning). Also, increasing the generalizability and reliability of wood species identification by using larger, more diverse and quality data sets or creating synthetic data sets is an important research topic.

### Funding
The author received no funding for this work.

### Competing Interests
The author declare that they have no competing interests.

### Author Contributions
- Halime Ergun conceived and designed the experiments, performed the experiments, analyzed the data, prepared figures and/or tables, authored or reviewed drafts of the article, and approved the final draft.

### Data Availability
The Matlab codes are available in the Supplemental File.

## Supplemental Information

Supplemental information for this article can be found online at http://dx.doi.org/10.7717/peerj.17021#supplemental-information.

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
