# Peer review of "Wood identification based on macroscopic images using deep and transfer learning approaches"

_PeerJ, doi:10.7717/peerj.17021_

## Round 0.1 · original submission · Major Revisions

Dear Authors, your article received four distinct reviews. Except for one of the reviewers who recommended rejection, and I do NOT agree with him/her, the other reviewers were very positive, and I agree entirely with them. Therefore, my decision, as the editor of your article, is to return to you for major revisions. Please, address all the questions and suggestions of the reviewers, improve your article, and send it to us, as soon as possible in order to speed up this process.

Reviewer 1 ·

Basic reporting

The introduction is clear, the writing is good, and the related literature is correct. However, there are few contributions in methods.

The species name should be written in italic style.

Experimental design

The transfer learning part is hard to understand, the author has not described the details of where and how to use the transfer learning.

Validity of the findings

The author has not introduced how to split the training dataset and test dataset. Although compared the training result in Table 1, it is not value for evaluating the model.

·

Basic reporting

Thanks for the submission. The paper studies the identification and classification of tree species using common deep neural networks with a diverse dataset. While applying deep learning approaches to tree species identification is not a novel idea, this paper expands on the number of models used for study as well as evaluation metrics and tested with potential optimization techniques such as lightweight models in the case of limited resources.

1. "Abstract" section:
a. From line 20: “...were trained using the transfer learning method.” In this case, what is the pre-trained network(or source model)? And the reasoning behind choosing the corresponding approach. After reading the following sections readers would know ImageNet pertain CNN model is used but it would be great to also include this information here.
b. From lines 28 to 29: “...Moreover, optimized approaches such as light network models offer the possibility of obtaining effective results even in situations with resource constraints…”. The description is a bit unclear as to how the optimization method is proposed and applied. Generally, when we talk about light models, we think of compression methods to shrink the model parameter sizes so they can run when memory and computing resources are limited. But based solely on this sentence it is difficult to tell which technique is used and why the author chose it. Would appreciate a bit more clarification on this.
c. From line 21: “... range of performance evaluation ” => evaluations.

2. "Introduction" section:
a. From line 58: “...were applied on..” on => to.
b. Great work on including the development of classification methods in tree species identification and how transfer learning was used in the previous work.

3. "Materials & Methods" section:
a. The introduction of transfer learning could use better wording. “(ML)” abbreviation could also be emitted. Although we reuse a model for a different task they are still related, and the main goal of using transfer learning is to boost network performance in a given task and to save time and resources. The first paragraph of the “Transfer Learning” section should reflect that.
b. From line 206: “..., ResNet-50 has 50 layers.” => , and ResNet-50.
c. From line 245: “and TN (true negative)” why does the last list item have the abbreviation to the front and “true negative” inside the parentheses?
d. What is the advantage of choosing MCC in binary classification? And when do we not use it?

Experimental design

The experiment design of this paper is very clear and easy to understand. A pre-trained CNN model is used in transfer learning tasks of all the other neural networks. Evaluation is done via metrics mentioned in the previous sections.

The limitations of types of transfer learning tasks and why the author chose the pre-trained method is also included in the “discussion” section.

a. From lines 320 - 321: “Such a replacement has been stated that it can help the model prevent dead neurons and learn better”. Would be great if the paper could include a few more sentences of the reasoning from the original paper.

Validity of the findings

The findings are reached by running various deep neural networks with the WOOD-AUTH dataset. The premises of the experiment design and the limitation of the identification and classification of tree species using the deep learning approach are identified by authors via domain knowledge and real-world examples.

Additional comments

Thanks for the opportunity to review the manuscript. Please address the aforementioned questions.

Reviewer 3 ·

Basic reporting

The manuscript presents a novel approach to wood identification using macroscopic images, leveraging deep and transfer learning techniques. While the topic is of significant interest, the current version of the manuscript reads more like a draft. The narrative, although clear in most parts, contains redundancies that could be streamlined for brevity and clarity. A comprehensive language and grammar review is recommended to enhance the manuscript's coherence and broader accessibility.

The introduction provides an overview of conventional wood identification techniques but could delve deeper. A more systematic comparison of these traditional methods would offer readers a richer context.

There are instances where claims are made without adequate justification or supporting evidence. i.e., the assertion that "Tree genus or species identification is required in fields such as plant taxonomy, timber trade, illegal logging, archaeology, conservation, art history, and criminology" needs references to substantiate the claim.

Experimental design

The Materials & Methods section could benefit from restructuring. Embedding code within the main text disrupts the narrative flow; consider moving it to supplementary materials. A table comparing models would be more reader-friendly than lengthy paragraphs.

When discussing evaluation metrics, it's more pragmatic to cite established definitions and emphasize any alterations or specific nuances pertinent to this research.

The manuscript could use more data augmentation techniques to increase the robustness and generalization of the model, such as rotation, flipping, cropping, scaling, etc.

Validity of the findings

The current 80-20 split for training and testing might not sufficiently validate the method's accuracy. Consider employing strategies like cross-validation or hold-out testing to gauge the model's performance on novel, independent datasets.

The manuscript could benefit from a more rigorous examination of potential confounders and biases. Discussions on data quality, model assumptions, image acquisition, and preprocessing would provide a more holistic view of the method's performance. Additionally, a comparative analysis with alternative methodologies achieving comparable or superior results would be insightful.

·

Basic reporting

This paper is an excellent and experimental summary of how to identify different wood species using deep learning technique. Material is understandable. While mentioning technical discussion / terms, wherever applicable sentences / paragraphs need to be restructured and logical transitions should be used to improve readability. Figures and tables should be added where they are being discussed, not just collectively at the end makes it challenging to follow. The article covers a lot of good content. Need to proofread before re-sending for grammatical errors.

Experimental design

The study fits in scope of the journal. However according to the PeerJ standards, submissions should clearly define the research question. There is no "research question" in a classical sense or gap in the research that is being addressed. There is no gap identified in literature being referenced ex. need of accurate deep learning techniques with less computational power on mobile applications. Methods and Approach are described in good details. Article has good findings, it fits more in criteria of “experimental findings” or “experimental study” than “research investigation”. Good efforts.

Validity of the findings

This article is missing how ShuffleNet / modified ShuffleNet achieved similar performance to more complex models while requiring less computational power. The conclusions must be appropriately stated and should be connected to the original question that is being investigated and should be limited to what is supported by the results. There is no sufficient comparison of various model performances and highlighting of the significant findings. The detailed analysis of the confusion matrix is not included, such as identifying the types that are most confused with each other. There is no information on how accurately each species were identified and its relations to the image numbers. Ex. Sajo type is the most misclassified type without any analysis. Sajo has image numbers 823 and 800 were train and 23 were test, that’s basically 97% train and 3% test. Cipres type is in similar range and no explanation how better Cipres did vs Sajo. In table 1 – ‘The species and image numbers used in this study’ there also needs to be one more column discussing how accurately this species was classified. While running experiments, was 80-20% or 70-30% tried for images that are closer to 800 ex. Sajo ,Cipres, Nogal cafetero rather than 800 fixed number of test images and what was the outcome? This study needs to discuss atleast 3 best deep learning architectures out of what was tried , that did better job specifying wood identification and how they compare against each other in terms of different parameters and then go for the best one in terms of need / requirement.

For more specific comment , refer "4. Additional Comments".

Additional comments

Abstract:

Abstract is nicely written, relevant, describing methodology in detail and results are quantified. It can use below improvements

1. Mention the source or nature of the images used. Are these publicly available datasets, or were they curated for this study?
2. You could mention that deep learning and machine learning are two types of artificial intelligence (AI). This would help readers who are not familiar with these terms. For example, "In recent years, the use of technologies such as artificial intelligence (AI), including deep learning and machine learning, has increased."
3. Add a bit more detail about the transfer learning method. Ex. fine-tuning a pre-trained model on a new dataset
4. What kind of customization of ShuffleNet model was done and what is its significance?
5. Make the conclusion concise, focusing on the unique contributions of the study and or implications for future research.



Introduction:

The introduction does an excellent job highlighting the importance of tree species recognition, It thoroughly presents classical methods and their challenges. This section provides a good overview of the state of the art in deep learning for tree species identification and classification. Blow are the ways this could be further improved
1. Flow might be enhanced with a slightly better organization of information. It’s more readable when this section begins with the importance of tree species recognition. Then moving into traditional methods, highlighting their challenges, before transitioning to image processing techniques.
2. After introducing the traditional methods and their challenges, provide a short transition that highlights the importance of technological solutions in the modern time before exploring into image processing techniques.
3. Terms "macroscopic" and "microscopic" are used before defining them in context. Add bit more to clarify this difference.
4. Discuss bit more - advantages and disadvantages of computer vision techniques in little more detail please.
5. Are there any specific challenges of wood species identification at the species level.
6. Add little more details into the limitations of traditional machine learning methods for wood species recognition.
7. Provide more information about the different types of image processing and machine learning techniques that have been used for wood species recognition. For example, mention of convolutional neural networks (CNNs), which have become the state-of-the-art for many image recognition tasks.
8. Are there any challenges of using image processing and machine learning techniques for wood species recognition. For example, one challenge is that wood samples can vary in appearance depending on their age, condition, and processing. Another challenge is that there is a lack of large, publicly available datasets of wood images.
9. Provide more information about the transfer learning method used by Sun et al. (2021), since good results were achieved with a small dataset ex. More detail on why ResNet50 was chosen, the benefits of LDA for feature enhancement, and the advantages of KNN for classification.
10. The transition from Sun et al. (2021) to the method developed by FabijaEska, Danek & Barniak (2021) is somewhat abrupt. It is not clear whether any connection between Sun et al. (2021) and FabijaEska (2021) or if they are two separate studies.
11. Provide more information about the extreme learning machine (ELM) algorithm used by Huang et al. (2021) as authors were able to achieve good results with a small sample size.
12. Provide more information about the pre-trained convolutional neural networks used by Elmas (2021), as good results were achieved with a large dataset.
13. Provide more information about the deep learning models used by Wu et al. (2021), as high accuracy is achieved with a dataset of 11 different wood species.
14. There's inconsistency in how different studies are presented. Some studies are introduced with their methods/results, while others only have results. It would be helpful to maintain a consistent format for better readability.
15. The frequent use of the pronoun "they" can make it unclear which researchers or studies are being referred to. This is especially noticeable when transitioning between different studies.
16. The transition between different studies and techniques could be smoother. Currently, the text moves quickly from one study to another, which might be hard to follow.
17. Add bit more introduction to tree species identification and classification and explain why it is an important problem.
18. Of all the studies mentioned, many of them are with relatively small datasets, add bit more to the challenge that how proposed models with small datasets would generalize for larger datasets.
19. Are there any advantages and disadvantages of deep learning techniques for tree species identification and classification, compared to traditional methods.
20. Provide more information on ShuffleNet architecture, and explain how it is different from other deep learning architectures.
21. Line 113 – Is this next paragraph, looks like unintended spacing

Materials and Methods:

Good description of the transfer learning, dataset, various architecture discussion and evaluation metrics. For better clarity consider below

1. Add if any specific ImageNet pre-trained CNN model/s were used.
2. Add the software version of MATLAB.

Dataset

3. Clarify the dataset resolution for better readability - "resolution of 640 × 480 pixels, containing 3.9 microns per pixel."
4. If there was any data augmentation techniques used, add briefly if it was used.
5. Add the limitation of this study, ex. model was trained on a dataset of tropical forest species, it may not perform as well on other types of trees.

Transfer Learning

6. The transfer learning section - you could start by explaining the importance or significance of understanding different architectures when considering transfer learning.
7. Could you add more specific examples of how transfer learning has been used in different domains, such as image classification, natural language processing, and computer vision.
8. Did author face any challenges of using transfer learning, such as negative transfer and the need to adapt the pre-trained model to the new dataset
9. The code snippet need to have proper context. Brief description of why it is added, may be provide additional comments for better clarity. How does this code relate to the broader discussion of transfer learning?
10. Architectural description needs to be organized. Each model's description has to be consistent in its format for an easy comparison. Consider using bullet points or separate subsections for more clarity if you believe this is appropriate for the context.
11. Streamline the descriptions of GoogLeNet and InceptionV3 to avoid redundancy.
12. What makes EfficientNet efficient? How do B0 to B7 differ?
13. When it comes to transfer learning how do these various architectures differ? Is there a reason to choose one over the other based on certain criteria? You may need to elaborate on this.
14. For this study, there is not much explanation on how the architecture selection was made from various deep learning architectures being discussed. What are all the different factors considered in this selection?
15. At the end of this section summarize the main points and possibly offer insights into which architectures might be most suitable for certain types of transfer learning tasks.

Evaluation Metrics

16. Begin by introducing the concepts of TP, FP, FN, and TN and then proceed to define the metrics. This will provide a more logical flow to the information.
17. F1-score vs. F1_score. It's important to maintain consistency.
18. The explanation of MCC is not clear for non-experts. Consider breaking it down further and provide a simple example wherever possible. For non-experts it’s not very clear from the text how MCC metrics / criterion is useful for imbalanced dataset.
19. Are there any specific examples of how the different evaluation metrics have been used in different domains, such as image classification, natural language processing, and computer vision.
20. Add description on how to use evaluation metrics to select the best model from set of different models? How the trade-offs between different evaluation metrics impact this selection?


Experimental Results:

Good overview of the different CNN models.

1. The section has lot information is presented in an intense manner. Consider breaking up the text into subsections, such as "Model Selection", "Transfer Learning Approach", "Model Modifications", and "Results", for better readability with smooth transition between ideas.
2. Out of so many different architectures only results and modifications of ShuffleNet v2 are deeply discussed without a summary of the other models' performances.
3. The changes and the reasons behind introducing modifications to ShuffleNet v2, could be more clearly articulated. Why were those specific layers changed, and what was the expected outcome – this information is crucial.
4. ShuffleNet v2 – is there any direct comparison before and after modifications.
5. Provide more information on the training process, such as the number of epochs used to train each model and the optimization algorithm used
6. Does this study considered the impact of different hyperparameters on the performance of the models? Can this be included if it was.
7. This section may certainly benefit from analysis of the results into more details, comparing model performances and highlighting significant findings.
8. Provide detailed analysis of the confusion matrix, such as identifying the types that are most confused with each other.
9. Line 315 and 316 - references "(To açar & Ergen, 2022)" and "(Zhang et al., 2018b)" are provided, but their relevance to the context needs to be clearly explained.
10. Line 348 - "The 9th type, Sajo type, was the most misclassified type" is presented without context and any deeper analysis. What makes this type challenging to classify? Are there specific characteristics of the images or similarities to other species that cause confusion? Consider discussing why this might be the case and any potential solutions.


Discussion:

This section is well-written and informative.

1. Line 353 - clarify what do you mean by "very specific tasks."
2. Line 357 - "Fine-tuning is performed" seem abrupt. This could be integrated more smoothly into the discussion rather than being abrupt. How was the fine-tuning performed?
3. Line 362 - "more layers are frozen or more weights are modified", need to elaborate on why this would be the case for large datasets.
4. If ShuffleNet was the primary model used, it should be discussed more regarding why this is a good choice for tree type identification and potential advantages over other models like MobileNet-v2.
5. If there's no standard dataset or metric, how that might have affected the results and interpretations?
6. What are potential applications of this study?
7. Conclude the discussion section by summarizing the main findings and highlighting the implications of this work for future research.
8. Is there enough research on ShuffleNet for its performance on tree type identification tasks.
9. Can some emerging trends in deep learning be applied to tree type identification, such as self-supervised learning and few-shot learning?

Conclusion:

Well-written conclusion that summarizes the main findings of the study and discusses the implications for future research.
1. Study mentions that deeper and more complex models performed better, but they also required more computational power and time. However, there is no discussion regarding the trade-off between performance and computational resources in more detail.
2. Line 420 – 423, these sentences are quite long and packed with information. Break them into shorter, clearer sentences to improve readability.
3. Conclusion mentions the importance of the dataset size, diversity, and quality for tree type identification with deep learning. It also mentions that the depth, number of layers, and number of parameters of deep learning architectures are important factors. However, there is no discussion on how these factors interact with each other. If the dataset is small or of poor quality, is it always the case that deeper and more complex models will perform better?
4. Conclusion mentions choice of transfer learning method is an important factor. It would be helpful to provide more specific examples of different transfer learning methods and how they can be used for tree type identification.
5. Are there any specific example of designing user-friendly and accessible applications and tools for tree type identification that can be used to provide education and awareness, and to benefit the conservation and sustainability of trees.
6. Be more specific about the limitations of the study. Ex. the study only evaluated a limited number of deep learning architectures and that the results may not be generalizable to other datasets.
7. Be more specific about the directions for future research. Like any specific feature extraction and transfer learning methods that could be compared in future studies.
8. There are no details about the ShuffleNet architecture and how it was used in the study. How ShuffleNet achieved similar performance to more complex models while requiring less computational power is unspecified from this conclusion as well as from the research article.

---

## Round 0.2 · Minor Revisions

Dear Authors,
Although the reviewers have clearly pointed out that this version was considerably improved, at least one reviewer still warned of issues that must be clarified and;or improved. Therefore, my decision is to send back to you as requiring Minor Revision.

Reviewer 1 ·

Basic reporting

This paper reviewed related literature well. Pointed out the difficulties of wood recognition and described the author's motivation. This research is useful and helpful in making wood recognition faster. The research is meaningful and the writing is clear.



1. suggestion: the paragraph of lines 101-104 should be merged into the previous paragraph.

2. The "ELM" should be written in full when first used.

Experimental design

The author compared 10 different models and found a candidate for wood species recognition. Choose models and evaluation metrics correctly. Some writing points should be considered.

1. line 390 "These technologies aim to make wood species identification faster....".
The paper did not explain if you improved the speed.


2. From line 208, the contribution uses max pooling instead of the mean pooling. The contribution is tiny. What about the difference in evaluation metrics between the max pooling and mean pooling?

Validity of the findings

1. line 491, "DenseNet201 model gave the best result with 97% accuracy and 0.96 MCC...."

The author has described the DenseNet201, but Table 3 shows the best model is Inception ResNetV2, why describe DenseNet201 not the best Inception in conclusions?


2. line 497, "The results show that the modified ShuffleNet model achieves similar performance...., less computation power..."

The best way is to use evidence to prove your conclusion.

Additional comments

1. Figure 1
10 species but only afford 9 images, the author should illustrate all of them. The sub-caption has only 8 species names, should write all 10 species names.

2. Figure 2
The author should label and explain the data in the bottom table in Figure 2.

·

Basic reporting

Thanks for submitting the reviewed version and providing your rebuttal comments.

1. "Abstract" section: the revised version addressed questions regarding the pre-trained models, and lightweight models(MobileNet and ShuffleNet with fewer parameters) with detailed explanation and evidence.

2. "Introduction" section: thanks for addressing the typos.

3. "Materials & Methods" section: authors have adjusted wording and some very minor description mistakes. The revised version also included reasons as well as the pros and cons of choosing MCC in binary classification.

Experimental design

The revised version has provided more background on choosing the replacement.

Validity of the findings

The findings are valid considering that it is a very specific topic studied by designing and running experiments with collected datasets. The models used are also open-sourced.

Additional comments

Thanks for the opportunity to review the revised manuscript. Based on the revised comments as well as data points I recommend this paper for publication.

Reviewer 3 ·

Basic reporting

The revised version of the manuscript has shown improvement in terms of clarity and coherence, but there are still areas that need further refinement.

The abstract is generally clear but can be made more concise. For example, instead of saying "Traditionally, this task is done by experts through observation and experience," you can simply state, "Traditionally, experts have identified forest types through observation and experience." or "While traditionally reliant on expert observation, recent advancement ..." to directly jump into the main topic. When discussing the models used, briefly explain what transfer learning is and why these specific models were chosen. This will help readers who may not be familiar with machine learning. Ensure that the abstract is easy to read and understand for a broad audience, including those who may not be experts in the field. Avoid overly technical jargon unless it's necessary and explain it when used.

In the introduction, while the authors touch upon traditional wood identification techniques, a more comprehensive comparison and critical evaluation of these methods would provide readers with a deeper context. Additionally, when introducing new viewpoints, like the statement, "Sometimes, identification at the genus level is sufficient for diagnosing the type of wood," it's crucial to follow up with a brief explanation and reference examples from previous journal articles to substantiate this claim. Moreover, the mention of "Microscopic method" and "macroscopic method" should be articulated more clearly and defined to ensure readers with varying backgrounds can understand their significance in wood identification.

Experimental design

Within the Materials & Methods section, the focus should shift away from introducing the concepts of transfer learning and CNN models and instead zero in on the fine-tuning process. Highlight specific alterations made to parameters or hyperparameters, and elucidate how these adjustments influenced the results. Enhancing clarity in this section can greatly benefit readers seeking to grasp the technical aspects of your methodology. Furthermore, incorporating a comprehensive workflow or framework of the model structure would contribute to a more lucid presentation.

Validity of the findings

Regarding the performance of ShuffleNet, a deeper dive into the reasons underpinning its superiority in the context of wood species identification is recommended. Provide an in-depth analysis with additional details and thorough model comparisons to elucidate the unique strengths of ShuffleNet. Additionally, it would be invaluable to explore the specific types of data for which ShuffleNet excels, demonstrating its superiority over other models and enriching the discussion.

·

Basic reporting

The V1 , which is fixed after feedback, looks much better. Thanks for fixing the review comments. This is a great work and would be very useful for wood identification study. Thank you for this research article. I am accepting this article.

Experimental design

NA

Validity of the findings

NA

---

## Round 0.3 · accepted · Accept

Dear Authors, I am very pleased to announce that I think you have made all the corrections and improvements suggested by all the reviewers, and now your manuscript is considerably improved and I am sure it will be accessed and used by many researchers all over the world. CONGRATULATIONS!